# Influence of Aortic Valve Stenosis and Wall Shear Stress on Platelets Function

**DOI:** 10.3390/jcm12196301

**Published:** 2023-09-29

**Authors:** Paweł Bańka, Maciej Wybraniec, Tomasz Bochenek, Bartosz Gruchlik, Aleksandra Burchacka, Andrzej Swinarew, Katarzyna Mizia-Stec

**Affiliations:** 1First Department of Cardiology, School of Medicine in Katowice, Medical University of Silesia, 40-635 Katowice, Poland; 2Faculty of Science and Technology, University of Silesia in Katowice, 40-007 Katowice, Poland; 3Department of Swimming and Water Rescue, Institute of Sport Science, The Jerzy Kukuczka Academy of Physical Education, 40-065 Katowice, Poland

**Keywords:** aortic stenosis, platelets function, hemodynamics, wall shear stress, valvular heart disease, antiplatelet therapy, cerebrovascular events

## Abstract

Aortic valve stenosis (AS) is a common heart valve disease in the elderly population, and its pathogenesis remains an interesting area of research. The degeneration of the aortic valve leaflets gradually progresses to valve sclerosis. The advanced phase is marked by the presence of extracellular fibrosis and calcification. Turbulent, accelerated blood flow generated by the stenotic valve causes excessive damage to the aortic wall. Elevated shear stress due to AS leads to the degradation of high-molecular weight multimers of von Willebrand factor, which may involve bleeding in the mucosal tissues. Conversely, elevated shear stress has been associated with the release of thrombin and the activation of platelets, even in individuals with acquired von Willebrand syndrome. Moreover, turbulent blood flow in the aorta may activate the endothelium and promote platelet adhesion and activation on the aortic valve surface. Platelets release a wide range of mediators, including lysophosphatidic acid, which have pro-osteogenic effects in AS. All of these interactions result in blood coagulation, fibrinolysis, and the hemostatic process. This review summarizes the current knowledge on high shear stress-induced hemostatic disorders, the influence of AS on platelets and antiplatelet therapy.

## 1. Introduction

Aortic valve stenosis (AS) is one of the most severe valvular diseases. According to the Euro Heart Survey, which examined the distribution of valvular heart disease in 25 European countries using a sample of approximately five thousand patients, AS is the most common valvular heart disease [1]. Its reported prevalence rises to nearly 10% in adults aged 80 years old and older, while the projected overall frequency reaches 3% in adults aged 75 years old and older [2]. As a result, AS has become a significant societal and economic challenge that is expected to advance in the near future. A comprehensive understanding of the underlying pathobiological mechanisms that drive AS is needed to improve prevention and treatment strategies. Aortic valve disease is a progressive chronic condition, which starts from minor fibrocalcific changes in the leaflets, known as aortic sclerosis, and progresses to more pronounced calcification, resulting in significant obstruction of left ventricular ejection in the later stages [3]. The leaflets of the aortic valve can slowly develop sclerosis, and the underlying pathological process is likely to be similar to that observed in atherosclerosis. This condition includes factors such as damage to the valve endothelium, lipid deposition, and an inflammatory response. Extracellular fibrosis and calcification characterize the advanced stage of the disease. The abnormal transformation of valvular interstitial cells (VICs) into osteoblasts may be the underlying concept for the ectopic ossification of the aortic valve [4]. Recent studies show that autotaxin (ATX) and lysophosphatidic acid (LPA) promote an osteogenic program in VICs [5]. It seems that activated platelets, which induce the release of autotaxin by VICs, may play a critical role in the pathogenesis of aortic valve stenosis. Although aortic stenosis is a common problem, its pathogenesis remains an interesting area of research. It should be noted that there is no prevention and conservative treatment available for patients with both early and advanced forms of the disease [3].

## 2. Mechanisms of Hemodynamics and Wall Shear Stress in Aortic Stenosis 

The main role of the aortic valve is to maintain appropriate pressure in the aorta during diastole of the heart and prevent regurgitation. A normal valve provides a laminar systolic blood flow. In the course of time, the leaflets of the aortic valve degenerate and fuse, which leads to the narrowing of the valve orifice. Blood flow through the aortic valve becomes turbulent as the stenosis progresses and the valve orifice area decreases [6]. The abovementioned changes influence the hemodynamics. Shear stress generated via abnormal blood flow involves a tangential force against the surface of the blood vessels [7]. This force increases with blood viscosity and flow velocity. Furthermore, shear stress affects not only the endothelium and vascular smooth muscle, but also impacts blood cells, leading to alterations in their functioning [8]. Changes occur in the anticoagulant capacity and permeability of the vascular endothelium, leading to conditions associated with atherosclerosis and promoting vascular calcification [4]. The effect of WSS was also demonstrated in coronary arteries, suggesting that WSS profiling may play an important role in plaque instability [9]. The inner layer of blood vessels, known as the intima, experiences constant exposure to hemodynamic pressures, such as vertical circulatory force and fluid shear stress. Turbulent, accelerated blood flow generated by the stenotic valve causes excessive damage of the aortic wall. Some researchers started to evaluate the fluid dynamics of aortic stenosis in many in vivo, in vitro, and in silico studies [10,11,12,13,14,15,16]. Blood flow velocity and valve pressure gradients were only examined in in vivo studies [11,12]. Early in vitro experiments used either straight circular tubes [10,15,16] with water as the medium, which could not adequately replicate the uneven characteristics of turbulence, particularly in regions such as the ascending aorta and aortic arch, or a curved circular tube with a fluid analogous to blood [13]. Yearwood et al. performed one of the first in vitro experiments using a human aortic model to study velocity, valvular pressure gradients, and turbulent axial stress [14]. The initial in silico studies were limited to the flow conditions of a 75% stenosis with viscous forces estimated using a Reynolds number up to 1000 and 2000 [15,16]. In the course of time, the studies’ conditions became close to those of real properties. Juhn et al. assessed the hemodynamic characteristics of mild, moderate, and severe AS by analyzing velocity patterns, laminar viscous wall shear, transvalvular pressure gradients and turbulent shear stress [17]. This evaluation was carried out using computational fluid dynamics (CFD) validated via experimental data. Three physiologically representative three-dimensional AS models covering ascending and descending aorta and aortic arch reflected more realistic fluid flow for the purpose of CFD (Figure 1). 

Juhn et al. performed series of flow simulations using the large eddy simulation (LES) technique, which is a mathematical model designed for tackling turbulence within the realm of computational fluid dynamics. To confirm the accuracy of the LES applied to the aortic stenosis model, they created a replica of the geometry using acrylic material. This step was performed to provide optical access for imaging with particle image velocimetry. A Newtonian blood substitute consisting of glycerol, water, and sodium iodide was used for this validation. The peak velocities were 2.0 m/s, 4.0 m/s, and 8.0 m/s for mild, moderate, and severe AS, respectively [17]. The jet flows of all AS became more skewed and narrower as the degree of stenosis increased. The areas of flow separation and stagnation increased with stenosis severity. In both moderate and severe AS, an asymmetric and irregular jet flow was observed as the fluid exited the stenotic area. The dimensions and form of the jet changed according to the severity of the AS, thus influencing the degree of asymmetric and irregular flow behavior. As the stenosis increased, the jet core moved closer to the posterior aortic wall and the jet length extended further into the aortic arch, causing the flow field to become more disturbed and chaotic. The extent of flow separation occurring immediately after the stenosis, along with its asymmetrical characteristics, expanded in proportion to the severity of AS. In cases of severe AS, the jet velocity shows significant asymmetry, with markedly elevated velocities (up to 8 m/s) adjacent to the posterior aortic wall and extending toward the aortic arch. Larger areas of flow separation and stagnation were observed around the jet near the anterior aortic wall. Transvalvular pressure gradients were measured at 14 mm of Hg for mild, 30 mm of Hg for moderate, and 113 mm of Hg for severe AS. The mean wall shear stress (WSS) and its standard deviations within the narrowed regions were as follows: 39.6 ± 12.4 Pa for mild AS, 104.7 ± 23.0 Pa for moderate AS, and 180.8 ± 95.2 Pa for severe AS (Table 1). In cases of both mild and moderate AS, the highest RSSmax values were noted in the regions of jet protrusion, but the highest RSSmax near to the anterior aortic wall was even more pronounced. Conversely, in severe AS, the RSSmax was most pronounced in the central part of the jet, coinciding with the zones of rapid velocity gradients and fluctuations.

Another interesting diagnostic tool for assessing aortic WSS is four-dimensional (4D) flow cardiovascular magnetic resonance (CMR). Blood flow dynamic parameters obtained with 4D flow CMR are expected to be an important indicator of left ventricular afterload, which may be related to remodeling or impaired left ventricular dilatation. Komoriyama and Kamiya et al. examined 32 patients with severe AS before and after TAVR using 4D flow CMR [18]. The study also included 12 control subjects who did not have significant aortic stenosis (AS) or aortic regurgitation. The assessment of blood flow patterns in the ascending aorta involved the use of streamlines, which represent the instantaneous velocity field at a specific temporal phase. Helical blood flow values were higher in the pre-TAVR group compared to the control group; however, there was no significant difference after TAVR. Moreover, the extent of vortical blood flow did not significantly differ between the pre-TAVR and post-TAVR groups compared to the control group. The average WSS and peak WSS in the entire ascending aorta were notably higher in both the pre-TAVR and post-TAVR groups compared to the control group. Patients after TAVR showed regional and global decreases in WSS, but it did not reach the level of the non-AS control subjects. Average WSS significantly decreased from 6.7 to 6.0 Pa and WSS peak from 52.0 to 47.5 Pa. TAVR reduced WSS in the right posterior wall of the middle ascending aorta and the anterior wall of the distal ascending aorta, which are strongly aggravated by the increased counterclockwise helical blood flow that passes through the stenotic aortic valve.

## 3. Influence of Shear Stress in Aortic Stenosis on Platelet Function

Aortic stenosis is associated with hemostatic abnormalities. Shear stress within the physiological range is required to maintain vascular function, but excessive WSS causes damage to blood vessels and blood cells. In healthy conditions, WSS promotes the production of nitric oxide, a potent vasodilator that helps to maintain normal blood flow and prevents platelet activation and adhesion. However, in the case of AS, the altered hemodynamics result in changes to WSS patterns, leading to endothelial dysfunction and subsequent platelet activation. Endothelial dysfunction triggered by AS-induced changes in WSS leads to the upregulation of adhesion molecules and the release of von Willebrand factor (vWF), a protein involved in platelet adhesion. Consequently, platelets become activated and undergo shape changes, promoting the exposure of glycoprotein receptors on their surface. These receptors facilitate the binding of platelets to vWF and collagen, initiating platelet aggregation and clot formation. 

Shear stresses in AS may involve pressure above 5 Pa. It is also related to type 2A acquired von Willebrand syndrome caused by the degradation of high-molecular weight multimers (HMWM) of vWF [19,20,21,22]. It can lead to gastrointestinal bleeding from angiodysplasia in the presence of aortic stenosis, i.e., so-called Heyde’s syndrome [23]. Shear stresses beyond normal physiological levels that arise within turbulent flows in AS prompt a structural alteration of von Willebrand factor (vWF) molecules. This alteration results in the exposure of binding sites (A1, A2, and A3) to ADAMTS-13, a vWF cleaving protease. The proteolysis of HMWM of vWF occurs at A2 cleavage site, leading to the formation of smaller multimers. Conversely, elevated shear stress has been associated with the release of thrombin and the activation of platelets, even in individuals with acquired von Willebrand syndrome [24]. Past studies showed the presence of tissue factor in leaflets of surgically explanted mineralized aortic valves [25,26].

Turbulent blood flow and shear stress may affect the endothelium and enhance platelet activation. Activated platelets release proinflammatory cytokines and chemokines that contribute to the development of inflammation within the aortic valve. This inflammatory response promotes the deposition of lipids and calcium, accelerating the calcification process and further narrowing the valve opening. Platelet-mediated inflammation also stimulates the production of reactive oxygen species, amplifying oxidative stress and vascular damage. Platelets can liberate mediators, including bioactive lipids such as LPA, which intensify local calcification (Figure 2).

Bouchareb et al. [5] proposed the hypothesis that activated platelets contribute to an osteogenic process and promote the progression of AS through the purinergic receptor P2Y1 (P2RY1)-glycoprotein IIb/IIIa (GPIIb/IIIa)-LPA pathway. They used a scanning electron microscope to analyze both the surfaces of the aorta-facing and ventricle-facing sections of mineralized aortic valves and then compared them to control aortic valves that were not mineralized. The activation of the endothelium was evident in regions situated on the side facing the aorta in mineralized valves compared to non-mineralized control valves [5]. The verification of endothelium activation in mineralized aortic valves was achieved through the assessment of vascular cell adhesion molecule 1 (VCAM1) levels using Western blotting. Microaggregates were also observed on the aorta-facing side of mineralized AVs, which consisted of fibrin-like material and platelets showing filipodia, a hallmark of activated platelets. All of the results suggest that the micro-aggregates of platelets at the surface of AVs may interact with VICs. To explore whether platelets stimulate calcification in aortic valves, Bouchareb et al. isolated human primary valvular interstitial cells (VICs) from non-mineralized aortic valves. They then cultured these cells in a medium that promotes mineralization, both with and without the addition of platelets activated by collagen. Adding collagen-activated platelets increased the mineralization of VIC cultures 1.2-fold [5]. Furthermore, the simultaneous treatment of VICs with both the osteogenic medium and activated platelets for a duration of 24 h resulted in an elevated expression of RUNX2 and BGLAP (Bone Gamma-Carboxyglutamate Protein) transcripts. In addition, activated platelets enhanced the activity of alkaline phosphatase, a marker of osteogenesis, by 2.2-fold. The above data confirm that platelets stimulate the osteogenic transition and mineralization of VIC cultures. Moreover, the researchers hypothesized that the mineralization of VIC cultures induced by platelets might be dependent on the autotaxin (ATX) released by VICs and its subsequent product, i.e., lysophosphatidic acid (LPA). The addition of an ATX inhibitor (HA130) prevented the mineralization of VIC cultures triggered by platelets [5]. In VICs, the addition of HA130 to the growth medium also prevented the release of BMP2. The experiment revealed that cells treated with HA130 released a smaller amount of BMP2, even in comparison to control VICs without platelets. This result can be associated with strong basal ATX activity in VICs, which could promote LPA-mediated BMP2 activation under baseline conditions [27,28,29]. In addition, the application of Ki16425, an inhibitor targeting LPA receptors 1–3, reversed the platelet-induced mineralization that was observed in VIC cultures [5]. The use of small interfering RNA specifically targeting ATX in VICs resulted in a significant reduction in ATX levels, as determined via ELISA. This intervention also stopped the platelet-triggered mineralization observed in cell cultures. The researchers quantified the release of LPA using a specific ELISA method that does not cross-react with associated lipids, such as phospholipids, lysophosphatidylcholine, and sphingosine-1-phosphate. When platelets were co-cultured with VICs, the release of LPA into the supernatant increased by a factor of 4.1. This effect was abolished by HA130, an inhibitor of ATX. The results suggest that the mineralization-promoting effect of platelets may be facilitated by LPA produced from ATX. Activated platelets produce mediators such as an adenosine diphosphate (ADP), while VICs contain ADP-responsive P2RY1 receptors [30]. Bouchareb et al. added the P2RY1 inhibitor MRS2279 to VICs cultures and measured ATX activity in the growth medium as a marker of enzyme secretion. Activated platelets triggered the release of ATX, and this process was stopped by MRS2279. Furthermore, the introduction of apyrase, an enzyme that reduces ADP levels, into the growth medium prevented the release of ATX induced by platelets in VICs. Similarly, exposing valvular interstitial cell (VIC) cultures to the P2RY1 agonist 2-methylthioadenosine diphosphate trisodium salt (2MeS-ADP) resulted in a 3.7-fold rise in ATX mRNA levels [5]. The impact was notably diminished through a significant reduction achieved using siRNA to downregulate P2RY1 in valvular interstitial cells (VICs). Furthermore, the suppression of P2RY1 in VICs resulted in a significant reduction in the platelet-induced mineralization observed in cell cultures. The treatment of VIC cultures with GR144053, which is an inhibitor of GPIIb/IIIa, negated the platelet-induced mineralization of cell culture. In the next phase of the study, the researchers quantified ATX activity in platelets freshly collected from both control subjects and patients with AS (mild to severe). Compared to controls without valvular disease, platelet-associated ATX activity increased four-fold in patients with AS [5]. Platelet-associated ATX activity was positively associated with the peak transaortic velocity and inversely associated with the aortic valve area. A number of different biological markers influence VICs in the progression of AS. A list of the most important interactions is presented in Table 2. 

A pivotal interaction between platelets and blood vessels involves vWF, a multimeric glycoprotein that plays a crucial role in maintaining hemostasis within the bloodstream. Megakaryocytes and endothelial cells are responsible for producing, storing, and releasing vWF in the form of ultra-large multimers. These multimers undergo cleavage upon release, facilitated by a-disintegrin and metalloprotease with a thrombospondin type-1 motif family (ADAMTS13). The vWF molecules range in size from single dimers to large multimers that circulate throughout the plasma. HMWM are composed of over 20 to 30 vWF subunits and exhibit optimal effectiveness in terms of hemostasis. Under conditions of elevated WSS, these HMWM undergo activation and elongation. This process enhances interactions with collagen and platelets, promoting their binding and activation. Additionally, this elongation exposes the binding domain for ADAMTS13. Ultimately, these HMWM undergo deactivation and are cleaved by circulating ADAMTS13, resulting in the generation of smaller and less effective multimers. Patients with AS have fibrotic and calcified valve leaflets that expose the circulating blood to high WSS. As a result, a significant number of circulating high molecular weight multimers (HMWM) undergo proteolysis. Consequently, patients with AS have lower circulating concentrations of these HMWM, and the concentrations are proportional to the degree of WSS. From a clinical perspective, this outcome leads to the observation that in severe AS, acquired vWF syndrome becomes an important concern. Additionally, the seriousness of AS, as determined based on the mean transvalvular gradient, exhibits a linear correlation with the reduction in circulating HMWM measured through blood tests. Thus, vWF has been demonstrated to be corelated with the severity of AS. Subsequent to clinical treatment, the concentration of HMWM instantly increases following transcatheter aortic valve replacements (TAVRs). This phenomenon is attributed to the normalization of shear stress levels, which consequently prevents proteolysis and the acute release of vWF from the endothelium. The levels of vWF are of significant importance in patients undergoing transcatheter aortic valve replacement (TAVR). A deficiency in vWF has been suggested to be associated with an elevated risk of bleeding during the procedure and proposed as a method for stratifying risk. However, conflicting data exist concerning the supporting evidence for these particular patients. For instance, Sedaghat et al. demonstrated that the levels of HMW vWF multimers were not associated with bleeding, but reduced vWF activity to antigen ratios was associated with bleeding [56]. Kibler et al. established a connection between vWF and late bleeding events following TAVR [57], while Grodecki et al. found that vWF analysis had no predictive value in patients undergoing TAVR [58]. More studies evaluating vWF and peri-operative and post-discharge bleeding are needed to further evaluate the utility of vWF analysis for determining bleeding risk. This aspect will play a pivotal role in finding a balance between the advantages and disadvantages of anticoagulation therapy, aiming to prevent valve thrombosis while considering the potential risk of bleeding that could affect patient outcomes.

## 4. Platelets Function in High- and Low-Gradient Aortic Stenosis 

AS can be classified into a high-gradient (HG) or low-gradient (LG) type based on the transvalvular pressure gradient. In HG AS, the obstruction of blood flow through the narrowed valve leads to increased pressure gradients across the aortic valve. The elevated shear stress and turbulent flow in HG AS can activate platelets, leading to shape change, increased surface expression of adhesive receptors, and enhanced platelet aggregation. HG AS has been associated with platelet hyper-reactivity and increased platelet–monocyte aggregation, which may contribute to inflammation and endothelial dysfunction. LG AS is characterized by a smaller pressure gradient across the aortic valve, often associated with reduced left ventricular function. In this condition, platelet function in AS may differ compared to HG AS. Reduced shear stress and turbulent flow in LG AS may result in less platelet activation compared to HG AS. However, platelet activation can still occur due to factors such as the platelet’s interaction with the damaged endothelium. Impaired left ventricular function in LG AS may lead to platelet dysfunction, including the impaired aggregation and secretion of platelet-derived bioactive molecules. Unfortunately, data specifically focused on platelet function in LG AS are limited. 

Knada et al. conducted a study to investigate the relationship between platelet activity and echocardiographic pressure gradients [59]. They studied individuals who had undergone surgical aortic valve replacement (SAVR). Peripheral blood samples were obtained at different time points: prior to surgery (T1), on the third (T2) and seventh (T3) postoperative days, and on the day of discharge (T4). Patients were assigned into groups with HG (peak pressure gradient > 100 mmHg) and LG (peak pressure gradient < 100 mmHg). At T1, there were no changes in four platelet factors in either group (PLT count, MPV, PDW, and P-LCR). Although there was a decrease at T2 in both groups, the PLT count increased more in the HG group than in the LG group at T4. In contrast, MPV, PDW and P-LCR were increased at T2 in both groups and diminished more at T4 in the group with HG [59]. In classifications based on the peak gradient, no differences in preoperative values were observed. However, there was a difference in postoperative platelet morphology, probably due to differences in shear stress. To establish a novel predictor to assess the magnitude of shear stress, plateletcrit (PCT) was calculated to represent the total volume of platelets in the blood both before and after surgery. There was no significant difference in PCT between HG and LG types. Consequently, patients were categorized into two groups based on whether they exhibited a substantial rise in plateletcrit (high PCT) or a minor increase in plateletcrit (low PCT). The researchers then compared preoperative platelet factors, PLT count, MPV, PDW, and P-LCR between these groups based on their respective rates of PCT. Notably, the group with a low rate of PCT increase demonstrated a significantly higher PLT count [59]. Conversely, PDW was significantly higher in the group with a higher PCT increasing rate. Neither MPV nor P-LCR underwent significant change. These findings indicate that elevated PCT, which should indicate high shear stress, leads to a reduced PLT count and an enlarged PDW prior to surgical aortic valve replacement (SAVR). Essentially, the impact of shear stress on platelets was more pronounced among patients with a lower PLT count and a larger PDW before surgery. WSS generated by AS activates platelets and influences their consumption and production (Figure 3). 

Intervention in symptomatic patients with high-gradient AS is recommended, regardless of the left ventricular ejection fraction (LVEF) [60]. However, the management of patients with low-flow low-gradient AS poses more problems. In patients with low-flow low-gradient AS with reduced ejection fraction, LV function usually improves after intervention if the reduction in LVEF mainly occurs due to excessive afterload. In contrast, this improvement is uncertain if the main cause of LVEF reduction is postinfarction scarring or cardiomyopathy. Intervention is recommended if stress echocardiography confirms the presence of severe aortic stenosis, while patients with pseudo-severe AS should be treated with conventional therapy. The presence flow reserve (an increase in the LV stroke volume of >20%) via a low-dose dobutamine test allows us to differentiate severe AS from pseudo-severe AS. On the other hand, higher procedural mortality is found among patients without flow reserve. Therapeutic decisions in such patients should take into account comorbidities, the severity of valve calcification, the extent of coronary artery lesions, and the revascularization options. Data on the natural course of low-flow, low-gradient AS with preserved ejection fraction, as well as the results of SAVR and TAVI in this group, remain controversial. An intervention should only be considered in patients with symptoms and significant valve stenosis after the effective treatment of the comorbidities. The prognosis of patients with normotensive, low-gradient AS with preserved ejection fraction is similar to that of moderate AS—systematic clinical and echocardiographic follow-up is recommended [60].

## 5. Platelets Function in Aortic Stenosis and Cerebrovascular Events 

AS is related to various risk factors associated with atherosclerosis, such as hypercholesterolemia, arterial hypertension, and advanced age. Conditions such as high left ventricular pressure with volume overload, hypertrophy, and remodeling can result in symptoms like sudden cardiac death, angina, syncope, and congestive heart failure. Moreover, AS is associated with an increased risk of atrial fibrillation and contributes to an elevated stroke risk among these individuals. The presence of a calcified aortic valve has been linked to occurrences of spontaneous cerebral embolisms. Additionally, aortic stenosis (AS) has been found to enhance the generation of thrombin, elevate platelet activation, and reduce fibrinolysis within the valve region. As a result, individuals with AS could have increased vulnerability to experiencing thromboembolic events and cardioembolic strokes. Understanding the relationship between AS, wall shear stress, and platelet function holds clinical relevance. Platelet dysfunction contributes to the development of thrombotic complications, which are often observed in patients with aortic valve stenosis.

Cardioembolic stroke is associated with worse outcomes, underscoring the need for effective prevention strategies in high-risk individuals, potentially including patients with AS. If AS is a strong and independent risk factor for ischemic stroke, antithrombotic therapy may be considered in some patients, even in the absence of other risk factors. The potential for thrombosis to develop on the surface of the native valve raises the risk of thromboembolic incidents for the patient. A pioneering study by Stein et al. in 1977 provided data concerning the native aortic valves of 19 patients with AS [61]. Among more than half of these patients, the researchers identified the presence of microthrombotic formations on the valve surface. Cardiovascular risk also increases in patients with aortic valve bioprosthesis. Tian et al. conducted a meta-analysis of the impact of leaflet thrombosis on hemodynamics and clinical outcomes after bioprosthetic aortic valve replacement. Twelve studies with 4820 patients were included. Leaflet thrombosis (LT) detected via multidetector computed tomography (CT) is common in bioprosthetic aortic valve replacement. The total prevalence of LT was 9.7% [62]. Patients with LT had a significantly higher mean pressure gradient compared to those without LT. Four studies reported the incidence of LT, showing an elevated risk of Major Adverse Cardiac and Cerebrovascular Events (MACCE) in these patients. When the results of these studies were combined, they indicated a heightened risk of adverse cerebrovascular events, particularly strokes, among the patients with LT.

According to a comprehensive Danish cohort study, patients with AS have an increased risk of ischemic stroke compared to controls of similar age and sex, even in the presence of atrial fibrillation [63]. Among all age groups, the incidence rates of ischemic stroke and the associated relative risks were notably higher in individuals with AS compared to controls. However, the relative risk was more pronounced in younger individuals. For patients with AS aged 65 years old or older, the risk of ischemic stroke significantly decreased after undergoing SAVR. In cases where atrial fibrillation was also present, the incidence of ischemic stroke was 1.5 times higher when aortic valve stenosis coexisted. Importantly, this elevated risk of ischemic stroke was substantially reduced within 6 months following SAVR, indicating a prothrombotic effect associated with the stenotic aortic valve.

## 6. Aortic Stenosis and Antiplatelet Therapy 

Antiplatelet therapy, commonly used in the management of cardiovascular diseases, aims to inhibit platelet activation and aggregation, thereby reducing the risk of thrombosis. While the use of antiplatelet agents in AS is not as established as it is in other cardiovascular conditions, several studies have explored their potential benefits. Aortic stenosis is associated with an increased risk of thrombotic events, such as strokes and myocardial infarctions. Antiplatelet therapy may play a role in reducing the risk of these complications by inhibiting platelet activation and reducing thrombus formation on the dysfunctional valve or the damaged endothelium. Focusing on platelet receptors and their signaling serves as the fundamental groundwork for current and emerging antiplatelet therapies. The majority of antiplatelet therapies employed in clinical practice are directed toward platelet receptors, such as cyclo-oxygenase (COX)-1, the P2Y12 receptor, or the integrin receptor. COX-1 inhibitors, such as aspirin, have been used to irreversibly bind COX-1 to decrease prothrombotic activity in platelets. P2Y12 receptor inhibitors prevent the P2Y12 activation that leads to platelet activation, including granule secretion and integrin activation through the PI3K/Akt pathway. Clopidogrel, prasugrel, and ticlopidine are classified as irreversible inhibitors of the P2Y12 receptor, while ticagrelor and cangrelor belong to the category of reversible P2Y12 receptor inhibitors. The significance of platelets in relation to leaflet thrombosis on bioprosthetic valves, as well as their subsequent effects on clinical results and valve durability, has garnered increasing attention due to their pivotal roles in thrombus formation. Subclinical bioprosthetic leaflet thrombosis and the associated restricted leaflet motion was first described by Makkar et al. in 2015 [64]. In some patients who underwent bioprosthetic AV replacement, restricted leaflet opening was evident via CT. The use of therapeutic anticoagulation in these patients can lower the incidence of leaflet restriction on follow-up CT. The most effective approach to antiplatelet and anticoagulation therapy after transcatheter aortic valve replacement (TAVR) for reducing the risk of stroke and preventing or treating leaflet thrombosis is still a subject of debate. The exact cause of thromboembolic complications and leaflet thrombosis after TAVR, whether primarily stemming from platelet-mediated or thrombin-mediated clot formation, remains uncertain. Current ACC/AHA guidelines advocate using dual antiplatelet therapy (aspirin and clopidogrel) for 6 months post-TAVR, followed by lifelong aspirin monotherapy in patients with no other indication for anticoagulation [65]. However, the ESC guidelines revealed a meta-analysis of three small RCTs, which showed a significant increase in major or life-threatening bleeding with dual antiplatelet therapy over acetylsalicylic acid at 30 days, with no difference in the ischemic outcomes [60]. Therefore, according to updated ESC guidelines, single antiplatelet therapy with acetylsalicylic acid is recommended. Additionally, three large, randomized trials have assessed clinical outcomes with various antiplatelet/anticoagulation strategies in patients undergoing TAVR. In the large GALILEO trial, patients following TAVR without another clinical indication for oral anticoagulation were randomly assigned to receive 3 months of treatment with either the direct factor Xa inhibitor rivaroxaban with aspirin or dual antiplatelet therapy with aspirin and clopidogrel. Treatment with rivaroxaban was associated with a higher risk of death and thromboembolic events, as well as major, disabling, or life-threatening bleeding [66]. In the recent POPULAR TAVR trial, patients without a clinical indication for anticoagulation were randomly assigned to receive either three months of aspirin alone or dual antiplatelet therapy consisting of aspirin and clopidogrel following TAVR. The trial found that the incidence of bleeding events and the composite outcome of bleeding or thromboembolic events after one year were lower in the group receiving aspirin monotherapy [67]. No clear association between restricted leaflet motion and stroke or transient ischemic attack was observed. The use of antiplatelet therapy in aortic stenosis remains an area of ongoing research, and there is currently no consensus regarding its routine use. The decision to initiate antiplatelet therapy should be individualized, considering factors such as the severity of AS, presence of comorbidities, and bleeding risk.

## 7. Conclusions

The influence of AS and wall shear stress on platelet function has emerged as an important area of study in cardiovascular research. Platelet activation and aggregation, triggered by altered hemodynamics in AS, contribute to thrombus formation, inflammation, and disease progression. The altered hemodynamics result in changes to wall shear stress patterns. Turbulent blood flow and shear stress activate the endothelium and promote the adhesion and activation of platelets. Activated platelets release proinflammatory cytokines and chemokines that contribute to the development of inflammation within the aortic valve. Platelets can liberate mediators, including bioactive lipids such as LPA, which intensify local calcification. Activated platelets promote an osteogenic program and the progression of AS through activation of the P2RY1—GPIIb/IIIa-LPA pathway. Antiplatelet therapy, commonly used in cardiovascular diseases, aims to prevent platelet activation and aggregation, reducing the risk of thrombotic events. In the context of aortic stenosis, the potential benefits of antiplatelet therapy have been investigated, although the optimal approach remains a topic of debate. Given the heterogeneity of AS patients and the complexity of thrombotic mechanisms, an individualized approach to antiplatelet therapy is crucial. Patient factors, such as age, comorbidities, bleeding risk, and concomitant medications, should be carefully considered when deciding on the best antiplatelet treatment. Understanding these mechanisms provides insights into potential therapeutic strategies aiming to modulate platelet function to reduce the burden of AS. Further research and clinical trials are necessary to develop targeted therapies and optimize management approaches, as well as to understand underlying molecular mechanisms linking altered wall shear stress to platelet dysfunction in AS.

## Figures and Tables

**Figure 1 jcm-12-06301-f001:**
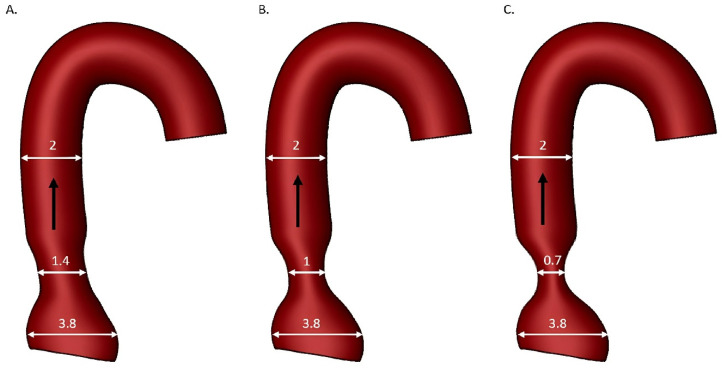
Hemodynamic models of aortic stenosis for (**A**) mild, (**B**) moderate, and (**C**) severe AS used in the experiment based on Juhn et al. [17]. Black arrow—direction of the stream flow, white arrows and numbers—diameter of the model expressed in centimeters.

**Figure 2 jcm-12-06301-f002:**
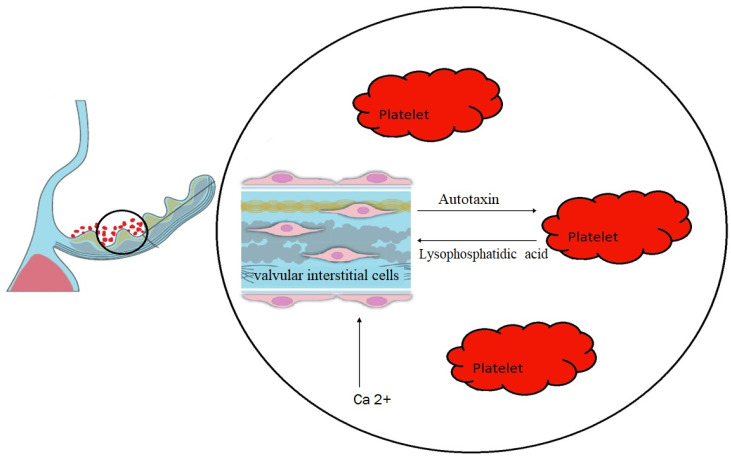
Interaction between platelets and valvular interstitial cells in aortic valve stenosis. The black circle symbolizes a close-up on a cross-section through the aortic valve leaflet. Platelets adhere from the aortic side to the valve leaflets. Inside the circle, the left side shows a cross section through the aortic valve tissue with its layers, from the top: valve endothelial cells (pink flat), lamina fibrosa, lamina spongiosa, lamina ventricularis and in the middle valvular interstitial cells (pink spindle-shaped). The arrows symbolize the interaction between the different molecules. Ca^2+^—calcium ions.

**Figure 3 jcm-12-06301-f003:**
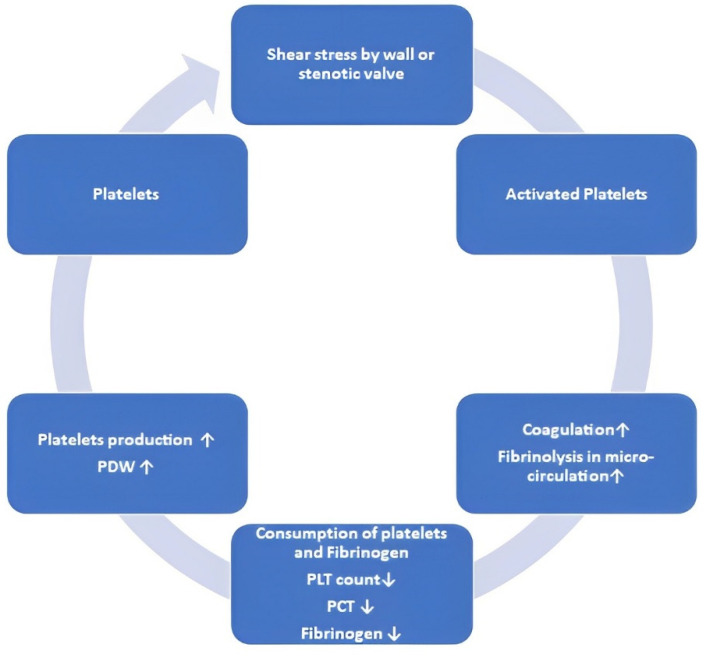
Mechanism of platelet changes in aortic valve stenosis. PCT—plateletcrit; PDW—platelet distribution width; PLT—platelet. white arrow up—increase or intensification; white arrow down—decrease or reduction.

**Table 1 jcm-12-06301-t001:** Flow simulation data in the models of AS [17]. AS—aortic stenosis, WSS—wall shear stress, RSSmax—maximal value of Reynolds shear stress.

Type of AS Model	Mild	Moderate	Severe
Peak velocities	2.0 m/s	4.0 m/s	8.0 m/s
Transvalvular pressure gradients	14 mmHg	30 mmHg	113 mmHg
Mean WSS	39.6 ± 12.4 Pa	104.7 ± 23.0 Pa	180.8 ± 95.2 Pa
RSSmax at centerline	260 Pa	490 Pa	2500 Pa

**Table 2 jcm-12-06301-t002:** List of molecules interfering with valvular interstitial cells in progression of AS based on Rutkovskiy et al. [31]. AKT—protein kinase B; BMP—bone morphogenetic protein; BSP—bone sialoprotein; GPIIb/IIIa—glycoprotein IIb/IIIa; HIF—hypoxia inducible factor; HOTAIR—HOX transcript antisense RNA; IFN—interferon; LPS—lipopolysaccharide; MMP—matrix metalloproteinase; NFκB—Nuclear factor kappa-light-chain-enhancer of activated B cells; OSP—osteopontin, PALMD—palmdelphin; P2Y2—purinoceptor 2; shRNA—short hairpin RNA; SORT—sortilin; STAT—signal transducer and activator of transcription; TGFβ—transforming growth factor β; TNF—tumor necrosis factor; TRAIL—TNF-related apoptosis-inducing ligand; VCAM—vascular cell adhesion molecule, V-LPP—VCAM-1 targeted lipopolyplexes.

Author, Year	Factor	Key Result, End Point
Osman, 2006 [32]	TGF β family cytokines and statins	Cytokines from the TGFβ family promote the differentiation of osteoblasts, whereas atorvastatin inhibits this process.
Osman, 2006 [33]	Adenosine triphosphate and statins	The activation of osteoblast differentiation is facilitated by adenosine triphosphate, but this effect is counteracted by atorvastatin.
Yang, 2009 [34]	LPS and peptidoglycan	Osteoblast differentiation is prompted by lipopolysaccharides (LPS) and peptidoglycan through the activation of toll-like receptors 2 and 4.
Yang, 2009 [35]	BMP 2	BMP2 triggers the early phases of osteoblast differentiation through both canonical and non-canonical pathways.
Yu, 2011 [36]	TNF α and BMP2	Tumor necrosis factor α exclusively triggers osteoblast differentiation in calcified VICs via BMP2 and NFkB signaling.
Carthy, 2012 [37]	Versican	VICs secrete versican in the wound assay; inhibiting its receptor CD44 leads to a reduction in stress fiber (αSMA) formation during VIC migration and inhibits collagen gel contraction.
Song, 2012 [38]	Biglycan	VICs derived from calcified valves exhibit elevated levels of biglycan expression. Biglycan, in turn, promotes osteoblast differentiation through the toll-like receptor 2 and ERK signaling pathways. The expression of biglycan and the calcification process are further stimulated by oxidized low-density lipopolysaccharides.
Zeng, 2012 [39]	LPS, toll-like receptor 4, and Notch	LPS activates an inflammatory phenotype through toll-like receptor 4 (TLR4). In calcified VICs, Notch1 enhances the responsiveness of toll-like receptor 4 to LPS through NFκB signaling.
Poggio, 2013 [40]	BMP 4	Bone morphogenetic protein 4 exclusively initiates osteoblast differentiation in non-calcified VICs, leading to higher levels of differentiation compared to osteogenic medium alone.
Zeng, 2013 [41]	LPS, Notch1	LPS stimulates the cleavage and nuclear translocation of the Notch1 intracellular domain, which subsequently triggers osteoblast differentiation via the activation of ERK and NFκB signaling pathways.
Nadlonek, 2013 [42]	Interleukin-1β	Interleukin-1β induces an inflammatory phenotype in VIC via NFκB.
Zhang, 2014 [43]	MicroRNA 30b	BMP2 initiates osteoblastic differentiation in VICs and suppresses the expression of microRNA 30b. MicroRNA 30b, in turn, inhibits osteoblastic differentiation and apoptosis.
Galeone, 2013 [44]	TNF-related apoptosis-inducing ligand (TRAIL)	Calcified VICs exhibit the presence of TRAIL receptors. The addition of TRAIL to the osteogenic medium enhances the formation of calcified nodules and promotes apoptosis.
El Husseini, 2014 [45]	AKT kinase and P2Y2 receptor	NFκB pathway is involved in inhibiting the expression of IL-6, which is a necessary factor for mineralization. Both AKT kinase and P2Y2 receptor activate this pathway, thereby suppressing IL-6 expression. Cells derived from *P2Y2*^−/−^ mice are prone to osteoblast differentiation.
Zhang, 2014 [46]	Transcription factor Twist	The osteogenic medium leads to the upregulation of Twist. This process leads to a decrease in the expression of other calcification-related genes. Conversely, the use of Twist siRNA induces osteoblast differentiation.
Carrion, 2014 [47]	Long noncoding RNA HOTAIR	Stretching downregulates HOTAIR through the Wnt signaling pathway. When siRNA is used to target HOTAIR, it leads to the upregulation of BMP2 and alkaline phosphatase expression.
Zeng, 2014 [48]	Oxidized low-density lipoproteins, LPS, and Notch1	Oxidized low-density lipoproteins enhance LPS-induced osteoblastic differentiation through the activation of NFκB and cleavage of Notch1.
Witt, 2014 [49]	Polyunsaturated fatty acids	Several polyunsaturated fatty acids can temporarily inhibit myofibroblast activation through the suppression of Rho kinase and ROCK kinase.
Song, 2014 [50]	Biglycan	Biglycan acts as a ligand for toll-like receptors 2 and 4, contributing to the activation of inflammation in VICs. This effect is mediated through NFκB and ERK pathways
Bouchareb, 2019 [5]	Autotaxin and lysophosphatidic acid	The release of autotaxin by VICs was induced by adenosine diphosphate derived from platelets. Autotaxin, in turn, bound to GPIIb/IIIa receptors on platelets, resulting in the generation of lysophosphatidic acid, which possesses pro-osteogenic properties.
Parra-Izquierdo, 2019 [51]	HIF-1α	HIF-1α activation via STAT1 in valve cells results in the proangiogenic, proinflammatory, and pro-osteogenic effects of IFN-γ
Wang, 2022 [52]	PALMD (Palmdelphin)	PALMD, a protein involved in myoblast differentiation, enhancing VIC osteogenic differentiation and inflammation through the activation of NF-κB.
Voicu, 2022 [53]	V-LPP/shRunx2 lipopolyplexes	VCAM-1 targeted lipopolyplexes, which downregulate the Runx2 gene and decrease the expression of osteogenic molecules OSP, BSP, and BMP-2 in VICs
Liu, 2022 [54]	MMP9	MMP9 expression was distinctly increased in AS, and its inhibition attenuated the calcification of valve interstitial cells by suppressing mitochondrial damage and oxidative stress.
Iqbal, 2023 [55]	Sortilin (SORT1)	Sortilin enhances fibrosis and calcification in aortic valve disease via the transformation of valvular interstitial cells into pathological phenotypes

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
