# Peer review of "Influence of Aortic Valve Stenosis and Wall Shear Stress on Platelets Function"

_jcm, 2023, doi:10.3390/jcm12196301_

Round 1

Reviewer 1 Report

The manuscript provides a comprehensive review of the impact of aortic valve stenosis (AS) on platelet function, focusing on the hemodynamic changes induced by AS and the resultant alterations in platelet behavior. The authors discuss the progression of AS, the contribution of high shear stress to hemostatic disorders, and the role of antiplatelet therapy in managing these challenges. The paper provides a valuable summary of current understanding in this area. Nonetheless, certain aspects of the study warrant further refinement.

1. Treatment strategies for high-gradient vs low-gradient aortic stenosis deserve a comparative discussion, to help guide clinical decision making.

2. When discussing antiplatelet therapy, the paper does not mention specific drugs, nor does it discuss the advantages, disadvantages, and applicability of various therapies. It is recommended that the authors supplement and expand this part.

3. The references cited in the paper are relatively old, and it is recommended that the authors update and cite some of the latest research to provide more comprehensive and cutting-edge information.

The authors should consider revising the manuscript to address the points raised in this review.

Reviewer 2 Report

The manuscript entitled “Influence of aortic valve stenosis and wall shear stress on plate-2 lets function” describes the interaction between wall shear stress in aortic stenosis and platelet function. The topic is interesting and the well-presented. Please, find below some comments. 

-       It would be interesting to describe data on wall shear stress after the correction of AS (either surgically or percutaneously). 

-       I would suggest to discuss also the only study investigating the role of WSS and gene expression in coronaries (Russo G, Pedicino D, Chiastra C, Vinci R, Lodi Rizzini M, Genuardi L, Sarraf M, d'Aiello A, Bologna M, Aurigemma C, Bonanni A, Bellantoni A, D'Ascenzo F, Ciampi P, Zambrano A, Mainardi L, Ponzo M, Severino A, Trani C, Massetti M, Gallo D, Migliavacca F, Maisano F, Lerman A, Morbiducci U, Burzotta F, Crea F, Liuzzo G. Coronary artery plaque rupture and erosion: Role of wall shear stress profiling and biological patterns in acute coronary syndromes. Int J Cardiol. 2023 Jan 1;370:356-365. doi: 10.1016/j.ijcard.2022.10.139. Epub 2022 Nov 5. PMID: 36343795.)

-       A figure summarizing the possible effects of WSS on molecules and platelet might make the paper more appealing to the readership. 
